# Integrating Network Pharmacology and Transcriptomic Strategies to Explore the Pharmacological Mechanism of Hydroxysafflor Yellow A in Delaying Liver Aging

**DOI:** 10.3390/ijms232214281

**Published:** 2022-11-18

**Authors:** Jie Kong, Siming Sun, Fei Min, Xingli Hu, Yuan Zhang, Yan Cheng, Haiyan Li, Xiaojie Wang, Xin Liu

**Affiliations:** 1College of Life Science, Engineering Research Center of the Chinese Ministry of Education for Bioreactor and Pharmaceutical Development, Jilin Agricultural University, Changchun 130118, China; 2College of Tropical Crops, Hainan University, Haikou 580228, China; 3School of Pharmaceutical Sciences, Wenzhou Medical University, Wenzhou 325035, China

**Keywords:** hydroxysafflor yellow A, liver aging, network pharmacology, transcriptomics

## Abstract

Aging affects the structure and function of the liver. Hydroxysafflor yellow A (HSYA) effectively improves liver aging (LA) in mice, but the potential mechanisms require further exploration. In this study, an integrated approach combining network pharmacology and transcriptomics was used to elucidate the potential mechanisms of HSYA delay of LA. The targets of HSYA were predicted using the PharmMapper, SwissTargetPrediction, and CTD databases, and the targets of LA were collected from the GeneCards database. An ontology (GO) analysis and a Kyoto Encyclopedia of Genes and Genomes (KEGG) pathway annotation of genes related to HSYA delay of LA were performed using the DAVID database, and Cytoscape software was used to construct an HSYA target pathway network. The BMKCloud platform was used to sequence mRNA from mouse liver tissue, screen differentially expressed genes (DEGs) that were altered by HSYA, and enrich their biological functions and signaling pathways through the OmicShare database. The results of the network pharmacology and transcriptomic analyses were combined. Then, quantitative real-time PCR (qRT-PCR) and Western blot experiments were used to further verify the prediction results. Finally, the interactions between HSYA and key targets were assessed by molecular docking. The results showed that 199 potentially targeted genes according to network pharmacology and 480 DEGs according to transcriptomics were involved in the effects of HSYA against LA. An integrated analysis revealed that four key targets, including *HSP90AA1*, *ATP2A1*, *NOS1* and *CRAT*, as well as their three related pathways (the calcium signaling pathway, estrogen signaling pathway and cGMP–PKG signaling pathway), were closely related to the therapeutic effects of HSYA. A gene and protein expression analysis revealed that HSYA significantly inhibited the expressions of HSP90AA1, ATP2A1 and NOS1 in the liver tissue of aging mice. The molecular docking results showed that HSYA had high affinities with the HSP90AA1, ATP2A1 and NOS1 targets. Our data demonstrate that HSYA may delay LA in mice by inhibiting the expressions of HSP90AA1, ATP2A1 and NOS1 and regulating the calcium signaling pathway, the estrogen signaling pathway, and the cGMP–PKG signaling pathway.

## 1. Introduction

Aging is inevitable in biological organisms, and it is a multidimensional process of organismal decline [1]. Although the liver is the most important detoxification organ in the body and has a strong regeneration capacity, aging induces a decrease in the number of hepatocytes and an increase in the volume of hepatocytes, which alters the structure and function of the liver [2,3]. Studies have shown that the induction of stressors, such as telomere dysfunction, DNA damage, and oxidative stress, can lead to the development of various liver diseases [4,5,6], such as hepatitis, cirrhosis, and liver cancer. All these diseases indicate that an aging liver is unable to cope with a variety of stimuli [7]. According to the World Health Organization (WHO), there are as many as 20 million patients with chronic liver disease or liver cancer worldwide, and 2 million people die each year due to liver failure, the highest portion of which are elderly people [8,9]. Currently, the aging process of the world population is accelerating, and it is urgent to explore the relationships between LA and various liver diseases. Therefore, the search for effective therapeutic approaches and the development of targeted drugs have important research value and clinical significance for delaying LA.

Safflower (*Carthamus tinctorius* L.) belongs to the Compositae family of herbs, which is a versatile economic crop and one of the most widely used traditional Chinese medicines [10]. The chemical composition of safflower is relatively complex, among which hydroxysafflor yellow A (HSYA) has the highest content and is the most pharmacologically researched [11]. HYSA is a compound with a monochalcone glycoside structure, a molecular formula of C_27_H_32_O_16_, and a molecular weight of 612 kDa. It is the main active ingredient in safflower [12]. HSYA has been proved to have various functions, such as antioxidant, anti-cardiovascular, anti-apoptotic, and anti-inflammatory effects [13,14]. In liver diseases, HSYA reduces liver fibrosis, regulates alcoholic liver injury, induces autophagy in hepatocellular carcinoma cells, inhibits angiogenesis of hepatocellular carcinoma, and prevents lung metastasis of hepatocellular carcinoma cells, indicating that HSYA is a potential hepatoprotective agent [15,16,17,18]. However, studies on HSYA in the treatment of liver aging have not been reported. In 2013, Kong et al. demonstrated for the first time that a topical application of HSYA could protect mouse skin from photoaging induced by ultraviolet radiation and revealed its mechanism of scavenging free radicals [19], indicating the potential biological activity of HSYA in improving skin and individual aging. Inspired by this study, our group previously treated D-galactose (D-gal)-induced aging mice with HSYA and found that an intraperitoneal injection of HSYA ameliorated D-gal-induced hepatic oxidative stress and replicative aging in mice [20]. These preliminary findings encouraged us to further explore the specific mechanisms by which HSYA delays LA.

With the rapid development of bioinformatics and systems biology, the combination of network pharmacology and transcriptomic analysis provides a new research model for revealing the complex mechanisms of action of drugs and even of compound Chinese herbal medicines [21,22]. Network pharmacology mainly breaks through the difficulties of studying the microscopic active ingredients of drugs, identifying drug targets through systematic “drug-target-disease” networks and analyzing and elaborating on the mechanisms of action of drugs from a holistic perspective [23,24]. In aging-related studies, Lan et al. explored the possible mechanisms of anti-aging of Astragalus via network pharmacology techniques [25]. Yang et al. discovered that the coenzyme Q10 exerted its anti-aging effects through signaling pathways such as PPAR and MAPK from the perspective of network pharmacology [26]. In recent years, the transcriptome has also been widely used in aging-related studies, such as using transcriptomics to explore the mechanism of the total flavones of epimedium flavonoids in delaying aging [27] and revealing the relationship between aging and inflammation based on the transcriptomics of acute kidney injury [28]. Single-cell transcription omics presented novel insights into aging and circadian processes [29], among others. Transcriptome analyses have the advantages of a wide detection range and deep data mining, and they are currently a high-tech method for aging research [30].

In this work, an integrative approach of transcriptomics and network pharmacology is used to investigate the biological mechanisms of HSYA delay of LA. We integrate the transcriptomic data from mouse liver tissue from previous experiments. Network pharmacology is employed to identify the potential targets of HSYA delay of LA. Subsequently, key molecular mechanisms are validated, and the interactions between HSYA and key targets are indicated through a molecular docking analysis. We provide a reference and experimental basis for HSYA research in the field of aging.

## 2. Results

### 2.1. Network Pharmacological Analysis

#### 2.1.1. Target Prediction and PPI Network Analysis of HSYA Delay of LA

A total of 8488 LA-related targets were collected from the GeneCards database, which intersected with 377 HSYA targets predicted from the SwissTargetPrediction, PharmMapper, and CTD databases; 199 targets were identified as potential targets of HSYA for delaying LA (Figure 1A). Then, the information on these 199 targets (Appendix A) was uploaded to the STRING database to obtain the corresponding protein interaction information (Figure 1B). The PPI network consisted of chemical components, target proteins, and differential gene proteins, including 137 nodes and 373 edges, where nodes represented proteins and edges represented relationships between proteins. According to the nature of network topology, there were 45 key nodes with degree and betweenness centrality values greater than the average (degree = 4.5, betweenness = 0.005974), accounting for 33.58% of the total number of nodes. The main central targets were genes such as MAPK1 (degree = 29), STAT3 (degree = 25), PIK3CA (degree = 25), HSP90AA1 (degree = 20), and EGFR (degree = 18) (Appendix A).

#### 2.1.2. Functional Enrichment Analysis of DEGs

To further explore the biological mechanisms of the 199 candidate targets in delaying LA, GO and KEGG pathway enrichment analyses of the candidate targets were performed using the DAVID database. In total, 468 GO items were identified by the GO analysis (Appendix A), and the top 66 GO items were screened (FDR < 0.01). The GO items related to biological processes mainly included signal transduction (GO:0007165), negative regulation of the apoptosis process (GO:0043066), protein phosphorylation (GO:0006468), and the oxidation reduction process (GO:0055114) (Figure 2A). Items related to molecular functions mainly included protein binding (GO:0005515) and ATP binding (GO:0005524) (Figure 2B). The entries related to cell composition were mainly cytosol (GO:0005829) and plasma membrane (GO:0005886) (Figure 2C). The KEGG enrichment analysis showed that 116 signal pathways were mainly enriched in the categories of “pathways in cancer” (hsa05200), “calcium signaling pathway” (hsa04020), “MAPK signaling pathway” (hsa04010), and “PI3K–Akt signaling pathway” (hsa04151) (Appendix A). The top 55 were screened according to FDR < 0.01 (Figure 2D). Using the disease analysis function of the NIMNT database, 199 potential targets of HSYA for delaying LA were uploaded, and possible indications of HSYA were enriched. After setting p.adjust < 0.001, 271 diseases were enriched (Appendix A), mainly including hepatitis, Alzheimer’s disease, and tauopathy (Figure 2E).

#### 2.1.3. HSYA Target Pathway Analysis

In this study, the 20 signaling pathways with the most enriched genes were selected, and the HSYA target pathway network model was constructed using the Merge APP of Cytoscape software. The center of the network was HSYA, and each target connected HSYA to the signal pathways (Figure 3). The results showed that, among the top 20 pathways by which HSYA delayed LA, the inflammatory mediator regulation of TRP channels, the cGMP-PKG signaling pathway, and the PI3K-Akt signaling pathway should be the focus of further research.

### 2.2. Transcriptomic Analysis

#### 2.2.1. Identification of DEGs

Figure 4A demonstrates the establishment of the D-gal-induced aging mouse model from previous experiments and the treatment of HSYA. In order to further explore the mechanism of HSYA in D-gal-induced aging mice, we performed an RNA sequencing analysis to obtain DEGs of samples from the control group, the D-gal group, and the D-gal group with HSYA. The results are shown in Figure 4B. Compared with the control group, 760 DEGs were highly expressed in the D-gal group, and HSYA could inhibit the high expressions of 338 DEGs. Meanwhile, compared with the control group, the D-gal group had 379 DEGs with low expressions, and HSYA could activate the low expressions of 142 DEGs. A total of 480 DEGs were regulated by HSYA (Figure 4C).

#### 2.2.2. GO and KEGG Pathway Enrichment Analyses

The analysis showed that 161 GO entries could be enriched from 338 highly expressed DEGs regulated by HSYA when *p* < 0.05 (Appendix A). There were 88 biological process items, including muscle contraction (GO:0006936), muscle filament sliding (GO:0030049), regulation of cardiac conduction (GO:1903779), etc.; there were 42 cell composition items, mainly including the Z disc (0030018), M band (GO:0031430), actin cytoskeleton (GO:0015629), etc.; and there were 31 items of molecular function, mainly including protein binding (GO:0005515), calcium ion binding (GO:0005509), actin binding (GO:0003779), etc. Figure 5A shows the GO enrichment analysis of the top 30 highly expressed DEGs by which HSYA delayed LA.

When *p* < 0.05, 28 GO entries could be enriched among the 142 low-expressed DEGs regulated by HSYA (Appendix A). Biological processes could be enriched for 19 items, mainly including oxidation–reduction process (GO:0055114), transmembrane transport (GO:0055085), lipid metabolic process (GO:0006629), etc.; cell composition was enriched for three items, including cytosol (GO:0005829), plasma membrane (GO:0005886), and endoplasmic reticulum membrane (GO:0005789). The molecular function was enriched for 6 items, mainly including L-ascorbic acid transporter activity (GO:0015229), nucleobase transmembrane transporter activity (GO:0015205), oxidoreductase activity (GO:0016491), etc. Figure 5B shows the GO enrichment analysis of the top 30 low-expressed DEGs for HSYA delay of LA.

A KEGG pathway enrichment analysis was further performed on HSYA-regulated DEGs (Appendix A). The results showed that HSYA regulated a total of 24 signaling pathways, in which 338 highly expressed DEGs could be enriched for 21 signaling pathways (*p* < 0.05), mainly including the calcium signaling pathway (hsa04020), the MAPK signaling pathway (hsa04010), and cardiac muscle contraction (hsa04260) (Figure 5C). A total of 142 low-expressed DEGs could be enriched through three signaling pathways (*p* < 0.05), including metabolic pathways (hsa01100), circadian rhythm (hsa04710), and biosynthesis of antibiotics (hsa01130) (Figure 5D).

#### 2.2.3. Analysis of KEGG Pathways Related to HSYA Delay of LA

Through the above pathway enrichment analysis, we found three signaling pathways with more DEGs that were also related to aging—namely, the calcium signaling pathway (hsa04020), the glucagon signaling pathway (hsa04922), and protein processing in the endoplasmic reticulum (hsa04141). The *SLC8A3*, *RYR1*, *MYLK2*, *PHKG1*, *TNNC1*, *TNNC2*, *ATP2A2*, *ATP2A1*, *NOS1*, *ADCY1*, *CACNA1S* and *SLC25A4* genes in the calcium signaling pathway were all upregulated in D-gal-induced aging mice, but HSYA could inhibit the expressions of these genes (Figure 6A). Seven highly expressed DEGs were enriched in the glucagon signaling pathway—namely, the *GYS1*, *LDHB*, *PHKG1*, *PGAM2*, *PYGM*, *PPARGC1A* and *CPT1B* genes (Figure 6B). HSYA could reverse the abnormal expressions of these seven DEGs. As shown in Figure 6C, the *DNAJB1*, *HSP90AA1*, *HSPH1*, *HSPA1L*, *DERL3*, *SYVN1*, *CRYAB*, *HSPA1B* and *HSPA1A* genes were involved in protein processing in the endoplasmic reticulum.

### 2.3. Preliminary Verification of HSYA Mechanism of Delaying Liver Aging

#### 2.3.1. Determination of Target Genes and Pathways

The 199 targets predicted by network pharmacology for HSYA delay of LA and the 480 DEGs sequenced from the transcriptome were intersected, and a total of four target genes were obtained—namely, *HSP90AA1*, *CRAT*, *ATP2A1* and *NOS1* (Figure 7A). The 116 pathways enriched by network pharmacology and the 24 pathways enriched by transcriptome sequencing were intersected, giving a total of 12 pathways obtained—namely the estrogen signaling pathway (hsa04915), the ErbB signaling pathway (hsa04012), the MAPK signaling pathway (hsa04010), the calcium signaling pathway (hsa04020), focal adhesion (hsa04510), fructose and mannose metabolism (hsa00051), the AMPK signaling pathway (hsa04152), the oxytocin signaling pathway (hsa04921), the cGMP-PKG signaling pathway (hsa04022), glycolysis and gluconeogenesis (hsa00010), the glucagon signaling pathway (hsa04922), and biosynthesis of amino acids (hsa01230) (Figure 7B).

#### 2.3.2. Mechanism of HSYA Delay of Liver Aging

The liver tissues of the control group, D-gal group, and D-gal group with HSYA mice were tested by qRT-PCR. The results showed that, compared with the control group, the liver tissue of mice in the D-gal group contained upregulated *HSP90AA1*, *ATP2A1* and *NOS1* genes at varying degrees (*p* < 0.05). After HSYA treatment, compared with the D-gal group, HSYA could significantly inhibit the expressions of *HSP90AA1*, *ATP2A1* and *NOS1* (*p* < 0.05), but the change in the expression of the *CRAT* gene was not significant (Figure 7C). At the same time, the expression levels of the HSP90AA1, ATP2A1, and NOS1 proteins in mouse liver tissue were detected by Western blotting, and the trend of the results was consistent with the results of the qRT-PCR detection (Figure 7D).

The common targets identified by network pharmacology and transcriptomic analysis of HSP90AA1, ATP2A1 and NOS1, acted on the estrogen signaling pathway (hsa04915), the calcium signaling pathway (hsa04020), and the cGMP-PKG signaling pathway (hsa04022) through 12 common pathways (Appendix A). We validated the activity of the cGMP-PKG signaling pathway (Figure 7E,F). Compared with the control group, the cGMP and PKG protein expression levels were decreased in the D-gal group (*p* < 0.05). The cGMP and PKG protein expression levels were increased in the D-gal group with HSYA compared to the D-gal group (*p* < 0.05). This indicated that the mechanism of HSYA delay of liver aging may be related to the above targets and signaling pathways.

#### 2.3.3. Molecular Docking

To further verify the reliability of the target prediction results, the main relevant targets were subjected to molecular docking verification. The interaction energy results obtained using Autodock vina 1.1.2 software for semiflexible docking are shown in Table 1. HSYA could effectively bind to the active pocket of the protein, and the binding energies to the three targets (NOS1, HSP90AA1 and ATP2A1) were all less than or equal to −7.0 kcal/mol, indicating that HSYA had strong docking activity with the core target genes [31]. The docking analysis of NOS1 showed that HSYA made hydrogen-bonding interactions with TRP414, ARG419, PHE589, TRP592, MET594, GLU597, and TRP711 at the active site. In the interaction with HSP90AA1, HSYA made hydrogen-bonding interactions with ASP54, LYS58, ASN106, and PHE138. In the interaction with ATP2A1, HSYA formed hydrogen-bonding interactions with GLU183, SER186, GLN202, ASP203, THR353, MET361, ARG560, ASP601, GLY626, ASP627, and ARG678. In addition, HSYA also formed a hydrophobic interaction with ALA440 (Figure 8).

## 3. Discussion

HSYA is one of the most important effective monomer molecules in safflower, with multitarget and multipathway pharmacological characteristics [32]. Combining network pharmacology and transcriptomics and rationally analyzing the results was helpful to further explore the core targets and mechanisms of action of HSYA delay of LA [33]. In this study, after using the NIMNT database to enrich the indications of HSYA, we found that hepatitis ranked first of 54 potential targets of HSYA, which is consistent with previous studies showing that HSYA can prevent inflammatory liver injury and liver fibrosis by inhibiting expressions of the inflammatory cytokines IL-1β, IL-6, and TNF-α [34,35]. Alzheimer’s disease and tauopathies were second only to hepatitis, indicating that HSYA may play an important role in the treatment of the above aging-related diseases.

In network pharmacology studies, two important signaling pathways of HSYA for delaying LA were also uncovered, namely, the regulation of TRP channels by inflammatory mediators and the PI3K-Akt signaling pathway. The TRP ion channel family is implicated in a variety of physiological and pathological processes, including calcium influx, regulation of the intracellular microenvironment, and maintenance of vascular function [36]. HSYA could increase calcium ions via TRPV4 channels, activate eNOS and its phosphorylation via protein kinase A, and promote NO production, thereby relaxing blood vessels, indicating that HSYA is a candidate for the treatment of cardiovascular diseases caused by aging [37]. The PI3K-Akt signaling pathway is closely related to cell proliferation and apoptosis and has been reported in aging diseases. Studies have shown that bitter melon and astragalus can inhibit D-gal-induced apoptosis by regulating the PI3K-AKT signaling pathway and increasing the activity of antioxidant enzymes in aging mice, thereby exerting anti-aging effects [38,39]. Sun et al. found that HSYA could inhibit human hepatocellular carcinoma cell proliferation and migration and promote apoptosis through the PI3K pathway [40]. Combining the results of this study with those of previous studies suggests that the regulation of TRP channels by inflammatory mediators and the PI3K-Akt signaling pathway may be potential pathways for HSYA to delay LA.

In the data from the transcriptome sequencing, two pathways related to HSYA delay of aging were likewise identified: the glucagon signaling pathway and protein processing in the endoplasmic reticulum. Glucagon raises blood glucose, exerting the opposite effect of insulin, and is a counter-regulatory hormone for insulin [41]. The screened *GYS1*, *PYGL* and *PHKG1* genes play crucial roles in the glycogenolysis step of the glucagon signaling pathway, and their overexpression destabilizes glycogenolysis and predisposes one to glycogen storage disease [42,43], which can lead to cirrhosis and liver fibrosis [44]. The results showed that HSYA could reverse the abnormal expressions of the seven DEGs that were enriched and could also inhibit the LDH changes caused by aging. Therefore, we hypothesized that HSYA could delay the liver damage caused by aging by maintaining the homeostasis of the glucagon signaling pathway.

The endoplasmic reticulum is an important site for protein synthesis in the body, and with the onset of aging, the function of the endoplasmic reticulum gradually declines and abnormal protein accumulates, which can cause the onset of endoplasmic-reticulum-mediated responses [45]. The heat shock protein family, led by HSP90AA1, plays an important role in this pathway and works to degrade misfolded proteins. Therefore, their aberrant expressions can lead to the accumulation of misfolded proteins, which affects protein processing and can induce polycystic liver disease [46]. Chen et al. demonstrated that HSYA could have a therapeutic effect on ischemic stroke patients by regulating autophagy and endoplasmic reticulum stress [47]. In this study, we found that nine highly expressed genes including *DNAJB1*, *HSP90AA1* and *HSPH1* were present in the liver tissue of aging mice, indicating an increased level of endoplasmic reticulum stress in the livers of aging mice. HSYA could regulate the expression of HSP family genes, indicating that HSYA may alleviate the endoplasmic reticulum stress level in aging mice and ensure the normal processing of proteins in the endoplasmic reticulum, thereby exerting an anti-aging effect.

By combining network pharmacology and a transcriptomic analysis, we finally identified four key targets (HSP90AA1, ATP2A1, NOS1 and CRAT1) and twelve signaling pathways. The target genes were mainly involved in the estrogen signaling pathway, the calcium signaling pathway, and the cGMP-PKG signaling pathway. In the present study, the *HSP90AA1*, *ATP2A1* and *NOS1* genes were inhibited by HSYA to delay LA, but the *CRAT1* gene only showed a decreasing trend. It has been suggested that HSYA may inhibit the high expression of messenger molecules, such as HSP90AA1, ATP2A1 and NOS1, in the liver, thereby protecting the balance of the pathway mediated by messenger molecules and maintaining the structural stability of specific target proteins [48,49,50], while the reason for the insignificant *CRAT1* needs further investigation.

The estrogen signaling pathway has two signaling pathways that function in different ways, nucleus- and membrane-initiated steroid signaling. In the nuclear pathway, estrogen binds to the receptors ERα and ERβ to form an ER–ligand complex, which exerts biological effects [51]. Li et al. found that disrupting the estrogen signaling pathway in animal models could lead to hepatic triglyceride accumulation and hepatic steatosis, suggesting an important role of estrogen in regulating hepatic lipid homeostasis [52]. Moreover, oxidative stress due to hepatic lipid peroxidation can also exacerbate damage to the liver [53,54]. It has been reported that aging-induced high expression of the *HSP90AA1* gene can interfere with the nuclear pathway and affect the binding of ERβ to estrogen, which in turn affects downstream transcription factors. In severe cases, it can also cause diseases such as estrogen resistance syndrome [55] and fatty liver [56]. In this study, we found that HSYA could inhibit the expression of HSP90AA1 in a mouse model of aging, suggesting that HSYA could delay the aging of mouse livers by inhibiting HSP90AA1 and regulating the estrogen signaling pathway in mouse liver tissue.

When the results of network pharmacology and the transcriptome were analyzed separately, the calcium signaling pathway was significantly enriched in both, suggesting that it may be the main regulatory pathway for LA treatment by HSYA. The calcium signaling pathway maintains the balance of intracellular calcium ions through both internal and external pathways, and it regulates the homeostasis of calcium ions through voltage-gated channels [57]. A study found that the *ATP2A1* and *NOS1* genes could regulate the calcium signaling pathway, and the high expression of ATP2A1 caused by aging could lead to interference with internal pathways so that the IP3R receptor or RYR receptor located in the endoplasmic reticulum and sarcoplasmic reticulum were in a state of regulation [58]. Additionally, incomplete calcium ion release led to elevated NOS1 [59], which ultimately affected other signaling pathways. Negash et al. showed that high expression of the calcium signaling pathway stimulated the activation of the NLRP3 inflammasome, which is one of the main causes of hepatitis [60]. In this study, the expression levels of the *ATP2A1* and *NOS1* genes in the livers of aging mice increased according to both predicted and experimental means, and HSYA significantly inhibited the expression levels of *ATP2A1* and *NOS1*, suggesting that HSYA could delay liver aging by inhibiting the *ATP2A1* and *NOS1* genes.

The cGMP-PKG signaling pathway is an important target pathway for cell signaling in the body and one of the pathways affected by the calcium signaling pathway. It is well-known that aging is a risk factor for the development of cardiovascular diseases, such as hypertension, coronary artery disease, and heart failure. The high expression of calcium transport subunits, such as ATP2A1, restores the decrease in cytoplasmic calcium concentration and myofilament sensitivity to calcium ions triggered by the specific activation of PKG1 isoforms, which leads to smooth muscle tension and an increased incidence of coronary artery disease [61]. cGMP, PKG, and PDE5 are also important regulatory proteins of this pathway. The cGMP and PKG proteins both play a role in regulating metabolism in the liver [62,63]. França et al. found that abnormal expression of the cGMP-PKG signaling pathway in the liver led to an increased incidence of hepatic encephalopathy [64]. Kuo et al. showed that KMUP-l protected the liver from oxidative-stress-induced damage by increasing vasodilatory NO and cGMP/PKG [65]. PDE5 inhibitors had good therapeutic potential for age-related cognitive decline or neurodegenerative disorders accompanied by memory performance decline [66]. We found that HSYA could increase the expression of cGMP and PKG proteins in the aging group, which is consistent with the above findings, suggesting that HSYA may also delay liver aging in mice by regulating the cGMP-PKG signaling pathway.

In conclusion, this study systematically elucidated the potential mechanism of HSYA delay of LA by combining network pharmacology and transcriptomic techniques. It was speculated that HSYA may delay LA in mice by inhibiting the expressions of HSP90AA1, ATP2A1 and NOS1 and by regulating the calcium signaling pathway, the estrogen signaling pathway, and the cGMP-PKG signaling pathway. This study provides a theoretical basis for the research and application of HSYA in the field of anti-aging.

## 4. Materials and Methods

### 4.1. Reagents and Materials

HSYA was purchased from Dalian Meilun Biotechnology Co., Ltd. (Liaoning, China; purity > 98%, powder). HSYA powder was dissolved in sodium chloride as a working solution. D-gal was provided by Sigma-Aldrich (St. Louis, MO, USA). Trizol reagent, Prime Script Master Mix, and SYBR-qRT-PCR Kits were obtained from TaKaRa Biotechnology Co., Ltd. (Dalian, China). BCA Protein Assay Kits were ordered from Bytec Biotech Co., Ltd. (Beijing, China). Anti-HAP90AA1 antibody, anti-ATP2A1 antibody, anti-NOS1 antibody, anti-PKG and anti-β-actin antibody were purchased from Wanlei Biological Technology Co., Ltd. (Shenyang, China). HRP-conjugated goat anti-rabbit IgG/HRP was obtained from Bioss Biotechnology Co., Ltd. (Beijing, China).

### 4.2. Collection of Targets for HSYA Delay of Liver Aging

We used the Chemical Book (https://www.chemicalbook.com/ (accessed on 15 December 2021)) platform to obtain the English name, chemical formula, structural formula, mol format, and SDF format of HSYA. We uploaded the information to the SwissTargetPrediction (http://www.swisstargetprediction.ch/ (accessed on 15 December 2021)), PharmMapper (http://lilabecust.cn/pharmmapper/ (accessed on 15 December 2021)) [67], and CTD (https://ctdbase.org/ (accessed on 15 December 2021)) [68] databases for target prediction. We integrated and deleted the HSYA duplicate genes obtained from the above three databases and determined the remaining as the final HSYA genes. GeneCards (https://www.genecards.org/ (accessed on 15 December 2021)) human gene database was used to annotate human genes and provide annotated prediction-related information [69]. We entered the keyword “Liver Aging” to obtain liver-aging-related genes and then input the liver aging target genes and HSYA target genes into Venny 2.1.0 (https://bioinfogp.cnb.csic.es/tools/venny/index.html/ (accessed on 16 December 2021)) [70] software and screened the intersection of the two as the HSYA target genes for delaying liver aging.

### 4.3. Network Construction and Analysis

The predicted targets were imported into UniProtKB (http://www.uniprot.org/ (accessed on 16 December 2021)) to standardize gene and protein names. We used the Search Tool for the Retrieval of Interacting Genes/Proteins 11.0 database (STRING, https://string-db.org/ (accessed on 16 December 2021)) to build a protein–protein interaction (PPI) network diagram [71]. The minimum required interaction score was set as 0.9. In the Database for Annotation, Visualization and Integrated Discovery platform (DAVID, https://david.ncifcrf.gov/ (accessed on 16 December 2021)), the candidate targets were analyzed for the enrichment of the GO and KEGG analyses. The condition was FDR < 0.01. The NIM molecular network tool database (NIMNT, http://www.idrug.net.cn/NIMNT/ (accessed on 16 December 2021)) was used to analyze the possible indications of HSYA targets for delaying LA [72]. Then, a HSYA target pathway network was constructed using Cytoscape 3.7.1 (National Institute of General Medical Sciences, San Diego, CA, USA).

### 4.4. Animal Experiments and Sample Collection

The biological samples for the animal experiments were derived from previous work [20] and were treated as follows: C57BL/6 male mice (18 ± 2 g) were purchased from Longsheng Laboratory Animal Technology Company (Changchun, China). All mice were placed in SPF-grade mouse incubators to maintain normal light and humidity with free access to water and food. The animal experiments were carried out in accordance with procedures approved by the Animal Care and Use Committee of Jilin Agricultural University (application approval number: 210726201100510465).

Thirty mice were randomly divided into three groups (n = 10): the control group, D-gal group, and D-gal group with HSYA. The mice in the control group were intraperitoneally injected with 0.1 mL/kg/d normal saline, the mice in the D-gal group were intraperitoneally injected with for modeling, and the mice in the D-gal group with HSYA were injected with 200 mg/kg/d D-galactose and 25 mg/kg/d HSYA. After 8 weeks, the mice were anesthetized with 2% sodium pentobarbital (0.2 mL/10 g; Siyuan Technology Co., Ltd., Beijing, China) and then euthanized. The liver tissues of the three groups of mice were collected and immediately stored in liquid nitrogen.

### 4.5. Transcriptomics Sequencing and Data Analysis

The kidney tissues were randomly selected from mice in the control, D-gal, and D-gal with HSYA groups (n = 3), and total RNA was isolated from the kidney tissues using Trizol Reagent (Takara Biomedical Technology, Beijing, China). The integrity and purity of the total RNA were analyzed with NanoDrop 2000 (Thermo Fisher Scientific, Wilmington, DE, USA). cDNA libraries were constructed, purified, and clustered using approximately 1 μg total RNA per sample. After cluster generation, the library preparations were sequenced on the Illumina HiSeq2500 platform (Bemec Biotechnology Co., Ltd., Wuhan, China). The whole RNA-Seq analysis was performed by Biomarker Technologies (Beijing). The raw data (in fastq format) were processed by removing reads that contained adapters and reads with low quality based on Q20 (error rate of base calling ≤ 1%) to obtain clean, high-quality data. The DEGs were screened with the BMKCloud platform (http://www.biocloud.net/ (accessed on 20 April 2020)); the *p*-value was adjusted by the false discovery rate (FDR), and *p* < 0.05 and fold change ≥ 2 were set to be statistically significant. GO functional enrichment and KEGG pathway analyses were performed on the OmicShare platform (https://www.omicshare.com/tools/ (accessed on 6 February 2021)) [73].

### 4.6. RNA Extraction and qRT-PCR Analysis

Total RNA was extracted from the liver tissues of mice. Agarose gel electrophoresis was used to measure the quality of extracted RNA. Reverse transcription of the total RNA (1 μg) was performed using a reverse transcriptase kit (Takara Biomedical Technology, Beijing, China). SYBR Green Mix (Takara Biomedical Technology, Beijing, China) was used for two-step real-time polymerase chain reactions. The analysis was performed on a StrataGene Mx3000P thermal cycler (Agilent Technologies, Santa Clara, CA, USA). The GADPH gene expression in each sample was used as an internal control. The fold change in the relative expression level was calculated by the 2^−ΔΔCT^ method (primer sequences are listed in Table 2) [74].

### 4.7. Western Blotting

Total protein was extracted from the liver tissues of the control, D-gal, and D-gal with HSYA groups of mice (n = 3) using a total protein extraction kit (Solarbio, Beijing, China). Each mixture was centrifuged at 12,000× *g* for 30 min at 4 °C, and the supernatants were collected. Protein concentrations were determined with a bicinchoninic acid (BCA) protein assay. Extracted proteins were separated on precast 12% sodium dodecyl sulfate polyacrylamide gel electrophoresis (SDS-PAGE) by electrophoresis and were transferred onto polyvinylidene fluoride (PVDF) membranes. The membranes were incubated with the designated primary antibodies overnight at 4 °C. Next, they were incubated with horse radish peroxidase (HRP)-conjugated goat anti-rabbit antibody (bs-0295G-HRP, 1:1000, Bioss, Beijing, China) for 1 h at room temperature. Signals were detected by enhanced chemiluminescence (ECL detection kit, Bio-rad, Hercules, CA, USA).

### 4.8. Enzyme-Linked Immunosorbent Assay (ELISA)

Commercial ELISA kits (Jiancheng Bioengineering Institute, Nanjing, China) were used to determine the cGMP levels in the liver tissue of each group of mice in accordance with the manufacturer’s instructions.

### 4.9. Molecular Docking

We downloaded HSYA data in SDF format from the Pubchem database (https://www.ncbi.nlm.nih.gov/pccompound (accessed on 28 December 2021)) and then imported them into Chemdraw 3D. We used the MM2 module for energy minimization and used the lowest energy advantage concept and saved the results for the mol2 file. We searched the Protein data bank (PDB, http://www.rcsb.org/pdb/home/home.do (accessed on 28 December 2021)) to find the protein structures of the downloaded targets, where the numbers of NOS1 and HSP90AA1 were 5uo2 and 6gr5, respectively. The structure of the ATP2A1 protein was not resolved in the PDB database, and the model was created by the AphaFold system (DeepMind Technologies Ltd, London, UK) and visualized with PyMOL 2.5 software (DeLano Scientific LLC, Palo Alto, CA, USA). After dehydration, hydrogenation, charge calculation, and non-polar hydrogen incorporation with Mgtools 1.5.6 software, the ligands and receptors were saved in pdbqt format. Molecular docking studies were performed using Autodock vina 1.1.2 software [75]. The coordinates of the target active pocket are listed in Appendix A, and Size_x = 20, size_y = 20, size_z = 20 for each target. The docking process was calculated with a genetic algorithm. All docking run options were default values. Finally, the docking results with the highest scores were visualized with PyMOL 2.5 software.

### 4.10. Statistical Analysis

The data were expressed as means ± standard deviation. Statistical analysis was performed with one-way analyses of variance and Tukey’s tests using GraphPad Prism 6.0 (GraphPad Software, Inc., La Jolla, CA, USA). * *p* < 0.05 means that the difference was significant after comparison based on the control group, while # *p* < 0.05 means that the difference was significant after comparison based on the D-gal group.

## Figures and Tables

**Figure 1 ijms-23-14281-f001:**
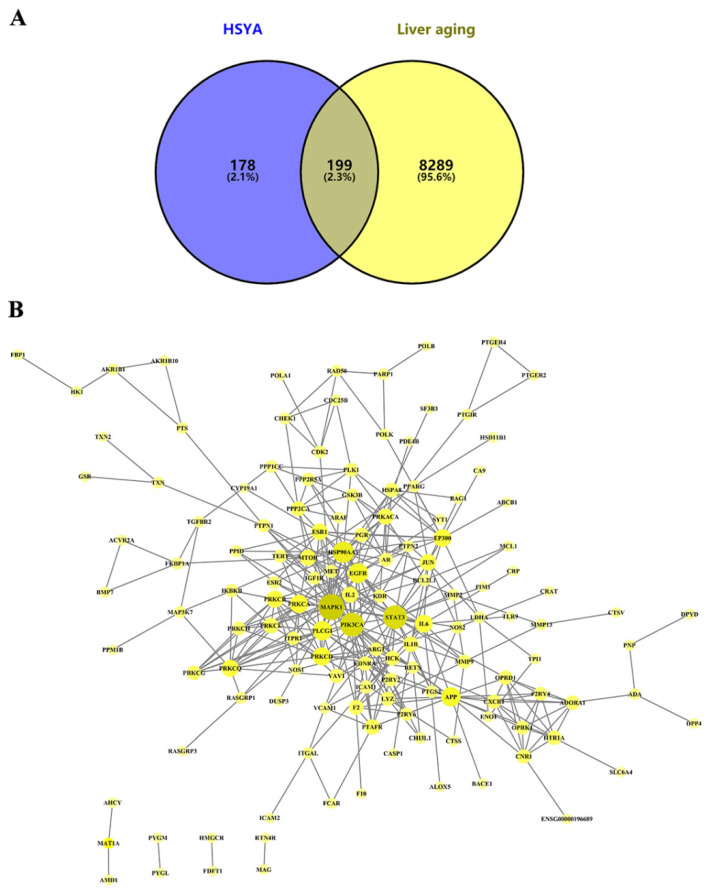
Potential targets of HSYA delay of LA. (**A**) Venn diagram of predicted targets of HSYA and LA. (**B**) PPI network analysis of relevant targets of HSYA delay of LA. Network nodes represent proteins, nodes represent potential proteins, and edges represent protein interactions. The sizes and colors of the circles were set according to the degree value of the node. The higher the degree value, the larger the protein and the darker the color, indicating that the protein could play a more central role. The color and thickness of the edges are positively correlated with the combined score of the proteins; darker colors and thicker edges represent higher importance.

**Figure 2 ijms-23-14281-f002:**
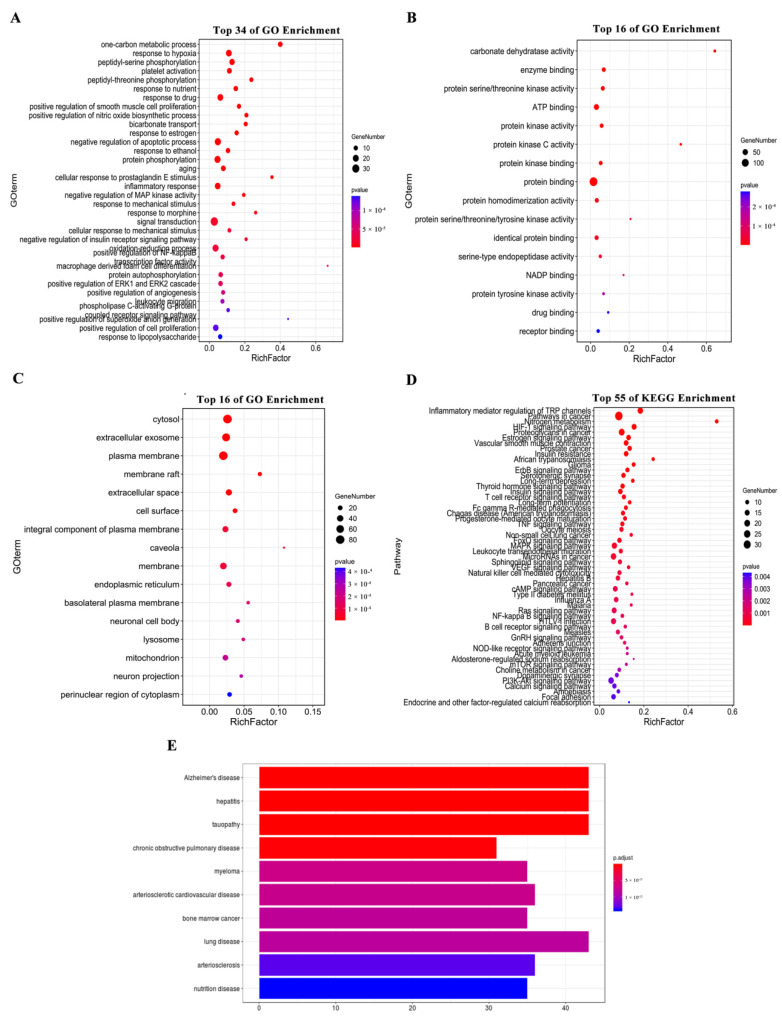
(**A**–**C**) GO enrichment analysis of 199 potential targets in terms of biological processes, cellular components, and molecular functions. (**D**) Enrichment analysis of KEGG pathways and (**E**) disease enrichment analysis of 199 potential targets.

**Figure 3 ijms-23-14281-f003:**
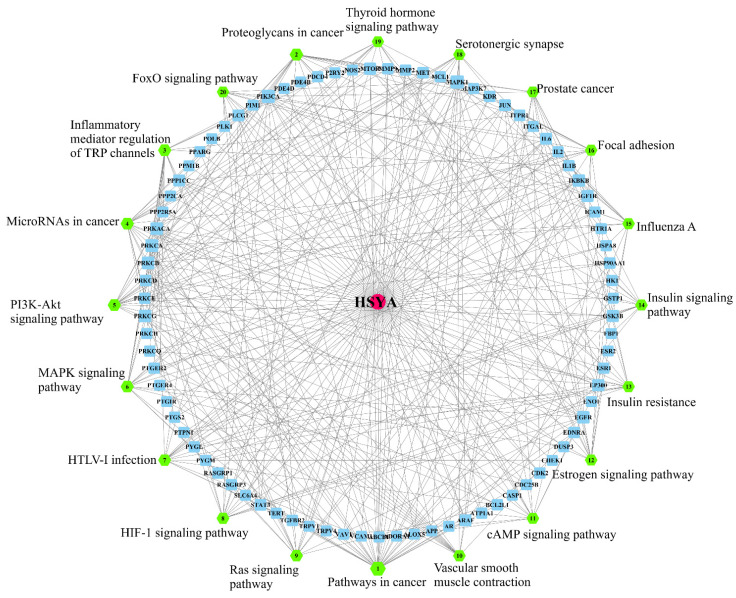
The target pathway network: red dots indicate the active ingredients; blue dots indicate the gene names; green dots indicate the signal pathways.

**Figure 4 ijms-23-14281-f004:**
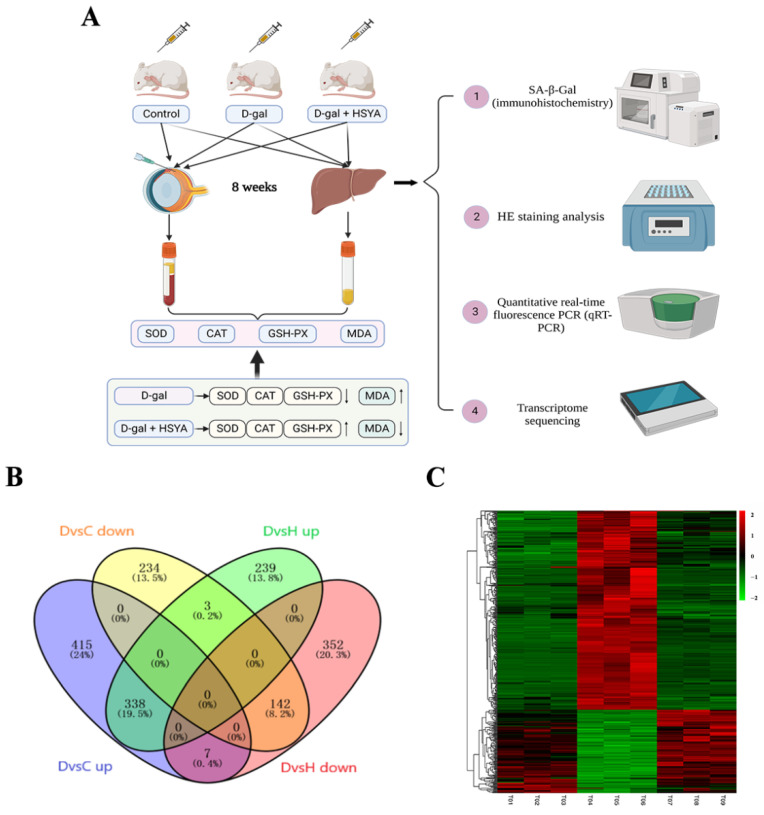
HSYA reversed the DEGs of D-gal-induced LA mice. (**A**) Protocol of the liver-aging model and treatment, as described in Materials and Methods. (**B**) Intersection of low-expressed DEGs (D-gal group vs. control group and D-gal group with HSYA vs. D-gal group), and intersection of highly expressed DEGs (D-gal group vs. control group and D-gal group with HSYA vs. D-gal group). C refers to the control group, D refers to the D-gal aging group, and H refers to the D-gal group with HSYA. (**C**) Heatmap analysis of low-expressed DEGs and highly expressed DEGs (total number of genes: 480).

**Figure 5 ijms-23-14281-f005:**
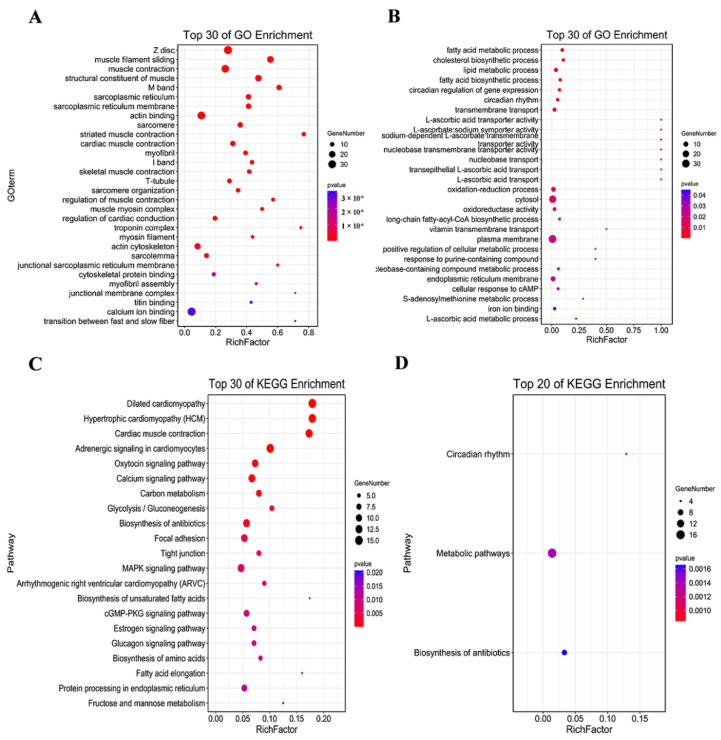
GO enrichment analysis of (**A**) highly expressed and (**B**) little-expressed DEGs for HSYA delay of LA. KEGG enrichment analysis of (**C**) highly expressed and (**D**) little-expressed DEGs for HSYA delay of LA. *p* <0.05.

**Figure 6 ijms-23-14281-f006:**
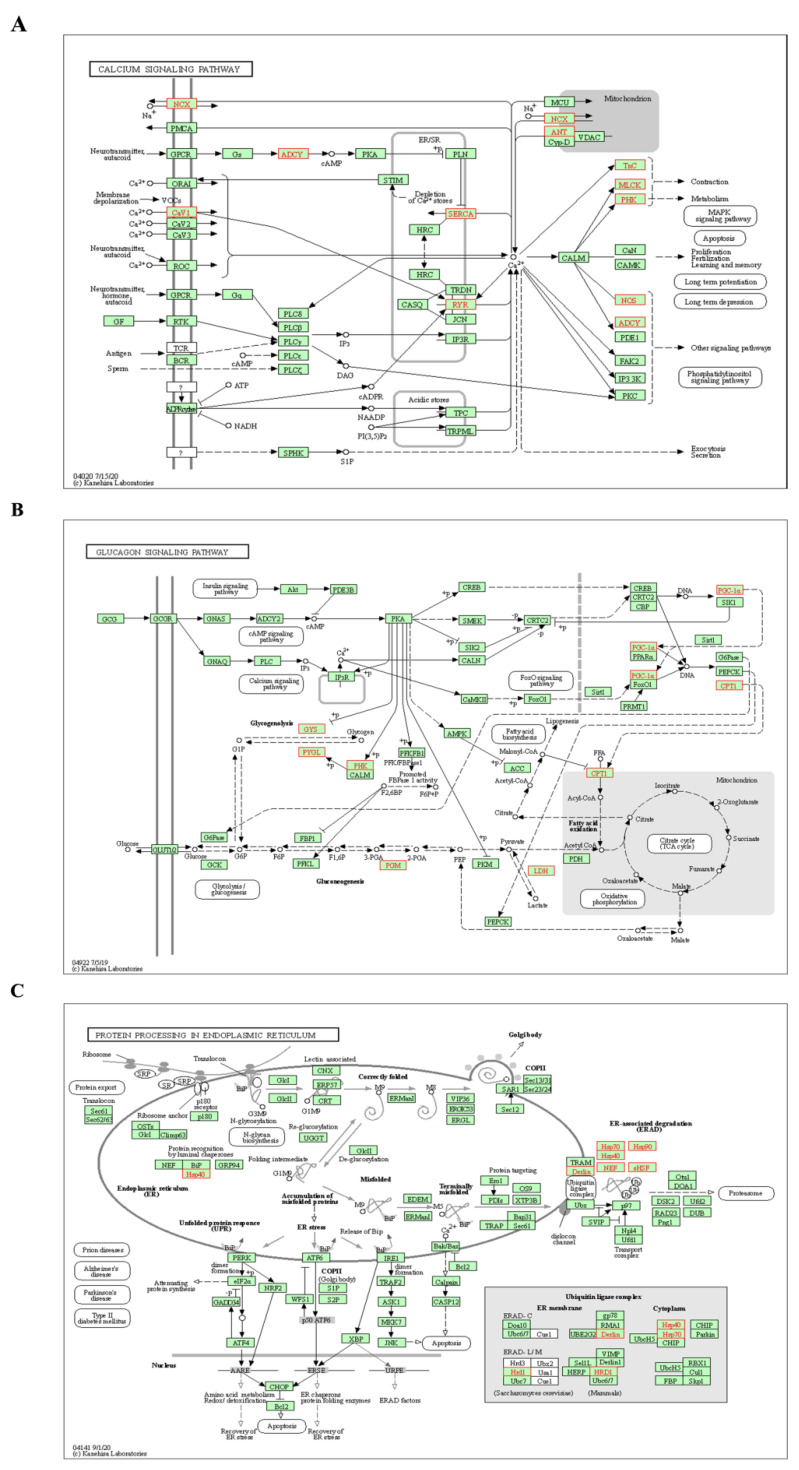
(**A**) Significantly enriched KEGG pathways in the calcium signaling pathway. (**B**) Significantly enriched KEGG pathways in the glucagon signaling pathway. (**C**) Significantly enriched KEGG pathways in protein processing in the endoplasmic reticulum. Upregulated DEGs are marked in red; pictures were drawn with KEGG Mapper (www.kegg.jp/kegg/tool/map_pathway2.html (accessed on 27 December 2021)).

**Figure 7 ijms-23-14281-f007:**
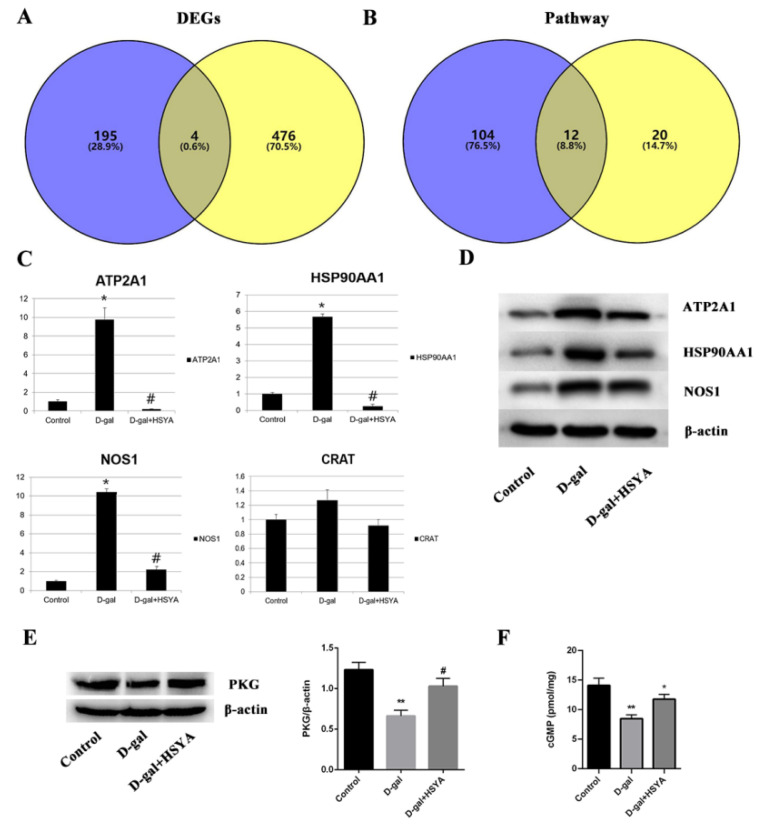
Preliminary verification of the mechanism of HSYA delay of LA. Venn diagram of (**A**) key targets and (**B**) signaling pathways combining network pharmacology prediction and transcriptomic analysis. qRT-PCR (**C**) and Western blot (**D**) to detect the expressions of four targets in mouse liver tissue. (**E**) PKG levels in the livers of mice from each group. (**F**) Expression of cGMP in the livers of mice was observed by ELISA. D-gal group compared with control group, * *p* < 0.05, ** *p* < 0.01; D-gal group with HSYA compared with D-gal group, # *p* < 0.05; data points represent means ± SD (n = 3); error bars indicate SD.

**Figure 8 ijms-23-14281-f008:**
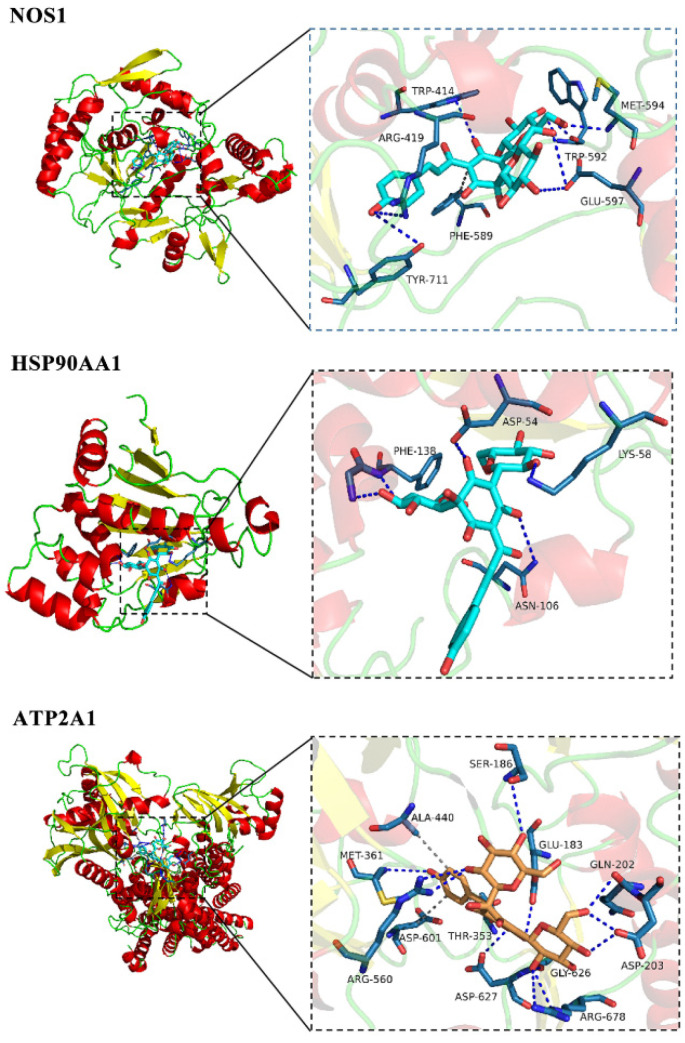
Docking results of HSYA with the targets.

**Table 1 ijms-23-14281-t001:** Component and target binding energies.

Target	PDB ID	Compound	Binding Affinity (kcal/mol)
NOS1	5uo2	HSYA	−8.8
HSP90AA1	6gr5	HSYA	−7.5
ATP2A1	none	HSYA	−7.0

**Table 2 ijms-23-14281-t002:** Primer information.

Gene	Primer Sequences	
HSP90AA1	Forward primer	ACCTTTGCCTTTCAGGCAGAA
	Reverse primer	CCGATGAATTGGAGATGAGCTC
ATP2A1	Forward primer	ACCTTTGCCTTTCAGGCAGAA
	Reverse primer	CCGATGAATTGGAGATGAGCTC
CRAT	Forward primer	AGAAGCTAAGCCCTGATGCCTT
	Reverse primer	GCGCAGAGAGGCACTTTCATAC
NOS1	Forward primer	TCCTAAATCCAGCCGATCGAC
	Reverse primer	CATTCACGAGGTCCTCATGGTT
GAPDH	Forward primer	AGGTCGGTGTGAACGGATTTG
	Reverse primer	GGGGTCGTTGATGGCAACA

## Data Availability

Not applicable.

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
