# Peer review of "Integrating Network Pharmacology and Transcriptomic Strategies to Explore the Pharmacological Mechanism of Hydroxysafflor Yellow A in Delaying Liver Aging"

_ijms, 2022, doi:10.3390/ijms232214281_

Round 1

Reviewer 1 Report

Dear authors, the manuscript entitled "Integrating network pharmacology and transcriptomics strategies to explore the pharmacological mechanism of hydroxysafflor yellow A in delaying liver aging" presents some interesting data. However, there are substantial flaws that have to be corrected.

In general, the English quality is too low.

-Several mistakes in the abstract.

-Introduction. There is some paragraph with unuseful information, being too much generalistic. You have to add relevant information in order to understand the results.

-Results there is some information that is discussion section.

-Discussion seems like a enumartion of each protein that you found and nothing more than explain what is their activity. Please, interpret your date, and discuss your own data with other manuscript. 

-Regarding the western blot and this is a main issue. There are four images for actin and three of them represent low protein levels in the first well, this means that the reduction levels found in the control group, in fact, desapear. So, in other words, in can not rely in the western blot results that you presented.

Reviewer 2 Report

Kong et al. took advantage of current bioinformatic pharmacologic prediction networks to predict the potential targets of HSYA in liver aging and explored these targets in a D-gal induced liver aging mice model majorly using transcriptomics and computational docking. Overall, this paper is well-structured, and the logic flow is sound. And the methodology used in this study can provide insights and contribute to the prediction and identification of uncharacterized targets of druggable compounds, especially herb extracts. However, the phenotypic characterization of the mice model and the role of the compound is not validated in detail and the pathway validation is limited. Thus, I would suggest publication after resolving the following issues.

Major issue:

1.     The authors used a D-gal-induced aging mice model to explore HSYA’s role in alleviating liver aging. However, there is no characterization of the animal model, as well as the validation of the efficacy of the HSYA. Without this information, any conclusion drawn from the transcriptome dataset is questionable. It is suggested to include the phenotype characterization of the animal model and the validation of HSYA’s role in this model.

Continue with this question, it would be preferred if the authors could show the intersected Venn diagram of DEGs identified in the transcriptome dataset (D-gal induced aging to control) with 8848 targets related to liver aging in the Gene Card database as a surrogate marker of how the whole dataset represents the mice model. If a very limited number of genes are co-altered in the animal model and the Gene Card database, the results observed by the authors -only identified 4 target genes and 12 associated pathways would be biased.

2.     Despite the fact that the authors kind of emphasizing potential pathways that contribute to the development of liver aging and the protective effects of HSYA. No validation of pathway activities of any emphasized pathways has been conducted. It would be preferred to show the activity of the cGMP-PKG signaling by western blot.

Minor issue:

1.     The language should be improved for this paper.

2.     In the methods, the animal study is conducted on 10 mice per group. However, it is not described the number per group used for transcriptome and western blot validation.

Round 2

Reviewer 2 Report

The modified version resolves all my concerns and I believe the paper is ready for publication.